# CD4+/CD45RO+: A Potential Biomarker of the Clinical Response to Glatiramer Acetate

**DOI:** 10.3390/cells8050456

**Published:** 2019-05-15

**Authors:** Martin Vališ, Lukáš Sobíšek, Oldřich Vyšata, Blanka Klímová, Ctirad Andrýs, Doris Vokurková, Jiří Masopust, Zbyšek Pavelek

**Affiliations:** 1Department of Neurology, Faculty of Medicine and University Hospital Hradec Králové, Charles University in Prague, Sokolská 581, 500 05 Hradec Králové, Czech Republic; martin.valis@fnhk.cz (M.V.); lukas.sobisek@yahoo.com (L.S.); oldrich.vysata@fnhk.cz (O.V.); Blanka.Klimova@uhk.cz (B.K.); jiri.masopust@fnhk.cz (J.M.); 2Department of Clinical Immunology and Allergology, University Hospital Hradec Králové, Sokolská 581, 500 05 Hradec Králové, Czech Republic; ctirad.andrys@fnhk.cz (C.A.); doris.vokurkova@fnhk.cz (D.V.)

**Keywords:** multiple sclerosis, glatiramer acetate, biomarker, CD4+/CD45RO+, patients, disease progression

## Abstract

**Background**: Glatiramer acetate (GA) is an effective treatment for the earliest stages of multiple sclerosis (MS)—clinically isolated syndrome (CIS) or clinically definite MS (CDMS). Objective: This study aims to determine the differences in the lymphocyte population (at baseline and the course of five years) between confirmed sustained progression (CSP) and non-CSP groups and to identify potential biomarkers among these parameters that can predict a positive response to the treatment. **Methods**: Twelve male and 60 female patients were included in the study. Peripheral blood samples were collected before and five years after treatment with GA. The authors compared lymphocyte parameters between the CSP and non-CSP groups by statistical analyses. Univariate and penalized logistic regression models were fitted to identify the best lymphocyte parameters at baseline and their combination for potential biomarkers. Subsequently, the ROC analysis was used to identify cut-offs for selected parameters. **Results**: The parameter CD4+/CD45RO+ was identified as the best single potential biomarker, demonstrating the ability to identify patients with CSP. Moreover, a combination of four lymphocyte parameters at baseline, relative lymphocyte counts, CD3+/CD69+, CD4+/CD45RO+, and CD4+/CD45RA+ab, was identified as a potential composite biomarker. This combination explains 23% of the variability in CSP, which is better than the best univariate parameter when compared to CD4+/CD45RO+ at baseline. **Conclusions**: The results suggest that other biomarkers can help monitor the conditions of patients and predict a favourable outcome.

## 1. Introduction

Multiple sclerosis (MS) is a chronic inflammatory demyelinating disease affecting the central nervous system. Glatiramer acetate (GA) is one of the basic drugs used for the treatment of multiple sclerosis and has shown a persistent and sufficient clinical efficacy by reducing the number of attacks and stabilizing and improving disability [1,2,3,4]. The direct mechanism of the GA effect is not known. GA treatment influences both innate and adaptive immunity: GA causes a shift from the activity of the Th1 subset to the activity of the Th2 subset. Through its action on APCs (antigen-presenting cells) such as dendritic cells and monocytes, GA changes the function of CD4+ and CD8+ T-lymphocytes. It possess a high affinity to MHC (major histocompatibility complex) II molecules on cells [5]. A meta-analysis of three randomized, double blind, placebo-controlled trials (n = 540) concluded that GA decreased the pooled adjusted annualized relapse rate (ARR) and decreased the accumulated disability (RR 0.6; 95% CI = 0.4–0.9; *p* = 0.02) [3]. Veugelers et al. [6] showed that the relative progression was significantly lower after starting GA (relative progression: 0.89; 95% CI = 0.81–0.97) compared with pretreatment rates. Ford et al. also concluded that MS patients who received GA for up to 15 years had reduced relapse rates, decreased disability progression and fewer transitions to SPMS; the mean duration of their disease was 22 years [7]. On the contrary, Munari et al. concluded that GA had no effect on MS disease progression [8]. GA is not sufficiently efficient for all patients with MS. Therefore, it is necessary to identify biomarkers that may predict a favourable response to the GA treatment.

In this study, individual populations of lymphocytes in MS patients (clinically isolated syndrome/CIS and relapsing-remitting multiple sclerosis/RR MS) were examined after five years of GA treatment. Patients were monitored for a period of five years. The aim of this study was to describe the differences in the lymphocyte population groups (at baseline and five-year changes) between patients with CSP (confirmed sustained progression) and those without CSP (non-CSP) and to identify among the baseline values of parameters potential biomarkers that may predict a positive response to the treatment.

## 2. Materials and Methods

### 2.1. Study Population

Participating subjects were all patients of the Department of Neurology at the University Hospital in Hradec Králové, Czech Republic. The study was conducted between 2008 and 2017. Twelve male patients and 60 female patients were included in this cohort study. Their age range was between 17 and 55 years. The mean age was 35 ± 9 years. All MS patients were of Caucasian background and met the McDonald criteria or revised McDonald criteria for RR MS [9,10]. The patients provided a proper medical history to obtain the necessary information about the expanded disability status scale (EDSS), disease duration and MS treatment history. Peripheral blood samples were collected before and five years after the treatment with GA. They were treated with subcutaneous injections of GA (20 mg each day).

The validation group for the ROC analysis consisted of 97 patients (25 male patients and 72 female patients) treated with interferon β-1b (subcutaneous injection, 250 µg every other day). Their ages ranged between 19 and 69 years. All MS patients were Caucasian and met the McDonald criteria or revised McDonald criteria for RR MS [11].

All participants provided written informed consent. The study protocol was approved by the Ethics Committee of the University Hospital Hradec Králové (reference number 201706S18P).

### 2.2. FACS Analysis

The blood samples of patients were collected from the antecubital fossa vein. The relative numbers of CD3+, CD4+, CD8+, CD19+, CD3-/CD16+56+, CD3+CD69+, CD3+CD25+, CD4+/CD45RA+, CD4+/CD45RO+, CD8+/CD38+, CD19+/CD5+, CD40 and CD40L lymphocytes were assessed by two-colour flow cytometry. For surface staining, 100 µL of blood was added to tubes containing 10 µL of fluorochrome-labelled mAbs, including fluorescein isothiocyanate (FITC)-conjugated anti-CD3 (clone UCHT1), anti-CD4 (clone 13B8.2), anti-CD45RA (clone ALB11), anti-CD8 (clone B9.11) and anti-CD19 (clone J3-119), as well as phycoerythrin (PE)-conjugated anti-CD25 (clone B1.49.9), anti-CD69 (clone TP1.55.3), anti-CD4 (13B8.2), anti-CD45RO (clone UCHL1), anti-CD38 (clone LS198-4-3), anti-CD5 (clone BL1a), anti-CD40 (clone MAB89) and anti-CD40L (clone TRAP-1), which were provided by Beckman Coulter (Miami, FL, USA). Class-matched isotype immunoglobulin FITC and PE-conjugated negative control monoclonal antibodies were added to individual tubes for all samples to identify nonspecific binding. All the antibodies used, their source information and dilution factors are summarized in Appendix A.

Thereafter, 100 µL of heparinised peripheral blood was mixed with the cocktail monoclonal antibody solution and incubated for 15 min at room temperature. A lysing solution (OptiLyse C, Beckman Coulter) was added, and the mixture was incubated for another 10 min. The flow cytometric evaluation was conducted with a Cytomics FC 500 cytometer (Beckman Coulter) equipped with a 15-mW air-cooled 488-nm argon laser and a 625-nm neon diode laser. All the data were assessed using the CXP Analysis Software (Beckman Coulter). A minimum of 10,000 events were obtained for each staining and supplied as a list mode. Multiple peripheral blood parameters were assessed in absolute and relative values. The 37 parameters included lymphocytes, CD4+ T-lymphocytes, CD8+ T-lymphocytes, CD 19+ B-lymphocytes, CD3-/CD16+/CD56+ natural killer cells, CD3+/CD69+ early activated T cells, CD5+ T cells, CD25+ cells, CD3+/CD25+ T cells, CD5+/CD19+ B cells, CD4+/CD45RA+ helper naïve T cells, CD4+/CD45RO+ helper memory T cells, CD8+/CD38+ T cells, CD69+ cells, CD40+ cells, and CD40L+ cells. The complete blood cell count was also determined. The data designated as ab represents an absolute count and the data without ab represent a relative count. The absolute values were calculated from the blood count, and the relative values were calculated as the percentage of lymphocytes.

### 2.3. Statistical Analysis

#### Statistical Processing

The patients were divided into two groups: CSP (confirmed sustained progression) and non-CSP group. The CSP group was defined by changes of the EDSS (1.0-point increase or greater if the EDSS result was more than 0.0 at baseline, or a 1.5-point increase or greater if the EDSS result was 0.0 at baseline). The non-CSP was defined by changes of less than a 1.0-point increase if the EDSS result was greater than 0.0 at baseline or less than a 1.5-point increase if the EDSS result was 0.0 at baseline.

First, we compared the lymphocyte populations (37 parameters) at baseline, at the end of the follow-up (same 37 parameters measured after five years) and their changes (37 absolute and 37 relative) near the end of the follow-up (last time point) between the CSP and non-CSP groups by running Welch’s one-way analysis of means (parametric *t*-test) for normally distributed parameters and Mann-Whitney test (nonparametric *t*-test) for non-normally distributed parameters. For the categorical parameter sex, a chi-square test of independence was used instead. The Lilliefors normality test was used to assess whether the parameters were normally or non-normally distributed. The effect size for all numerical parameters (except for sex) was assessed by Cohen’s d, which is the standardized difference between group means. Cramer’s V is reported for sex.

The association strength between the lymphocyte parameters and progression of EDSS were investigated by estimating a series of univariate logistic regression models for each explanatory variable (parameter) of 148 (4 × 37), where the binary response variable was the patient’s group (non-CSP or CSP). Thereafter, the statistical significance of parameters was validated by an adjusted (multivariate) logistic regression model with added covariates, sex, age and EDSS at baseline, to reduce the (latent) possible effect of these covariates on the response (dependent) random variable (non-CSP or CSP). The data were screened for the presence of the outliers for the regression modelling. We defined an outlier as a value that lies three times the interquartile range away from the median.

To identify the composite predictor (set of biomarkers) of clinical progression, we fitted two multivariate logistic regression models of CSP. In both models, only baseline lymphocyte values were entered as explanatory variables. Strongly articulated explanatory variables (multi-collinearity) are not appropriate to combine in one model. Multi-collinearity was determined to exist if the absolute value of the Spearman’s correlation coefficient exceeded 0.8. Strong (pair) dependence has been identified in a number of lymphocytic pairs. For this reason, the best combinations of explanatory variables use multivariate models (penalized logistic regression) that take into account multi-collinearity and multivariate models that do not have highly dependent variables together.

First, a penalized (lasso) logistic regression model was fitted via the penalized maximum likelihood with a 10-fold cross-validation. This model included all individual 37 parameters (their values at baseline). Subsequently, the identified combination (set), which was not strongly mutually correlated, of parameters entered the multivariate logistic regression model to compare the goodness of fit of this combination with each parameter independently and with further combinations (the second multivariate model). Not strongly correlated pair of parameters were defined as when the absolute value of their correlation coefficient was lower than 0.9. The choice of one explanatory parameter from the strongly correlated pair was made with the objective criterion Nagelkerke pseudo R^2^ (R2).

The second (fitted) multivariate model contained all statistically significant parameters (values at baseline) from the univariate models. The same selection of not strongly correlated parameters was applied as described above. The goodness of fit of each univariate and multivariate logistic regression model (strength of association) was evaluated by R2.

To control the false discovery rate, the Benjamini-Hochberg procedure using *p* < 0.05 as threshold of statistical significance was applied. The calculations were performed in the statistical system R (r-project.org) to obtain the penalized logistic regression model fit; the package *glmnet* was used [12,13].

We run ROC analysis to identify cut-offs (thresholds) for the four parameters, relative lymphocyte counts, and CD3+/CD69+, CD4+/CD45RO+, and CD4+/CD45RA+ab levels at baseline, which were chosen by the penalized logistic regression as potential biomarkers for CSP. The closest top left point lying on a built ROC curve was chosen as a cut-off for CSP.

## 3. Results

### 3.1. Statistical Comparison of CSP vs. Non-CSP Groups (GA Administered)

The group of patients treated with GA consisted of 72 patients (12 male and 60 female patients). Group A (CSP) consisted of 32 patients (6 male and 26 female patients), group B (non-CSP) consisted of 40 patients (6 male and 34 female patients). Therefore, there were 4.33 times more women than men in group A and 5.67 times more women than men in group B. The difference between the groups and the proportion of women was not statistically significant (*p* value 0.916) at a 5% significance level. The groups were not significantly different in age. The mean age (standard deviation) of patients in groups differed on average by 2.2 years (A 36.2 (8.09), and B 34.0 (9.9)). This is not a statistically significant difference (*p* value = 0.309). Groups with a statistically significant difference (*p* < 0.438) did not differ in basal EDSS (median A 2.0 vs. B 1.5).

Of the lymphocyte characteristics, the differences between A and B were statistically significant (at the 5% level of significance) for baseline CD4+/CD45RO+ (mean A 25.1 vs. B 19.9; *p* value < 0.001; effect size = 0.9), relative lymphocyte counts (26.3 vs. 32.2; 0.005; 0.69) and CD3+/CD69+ab (0.07 vs. 0.05; 0.043; 0.32).

In the fifth year, the groups differed significantly in absolute lymphocyte counts (2.0 vs. 2.11; 0.016; 0.59), CD3ab (1.5 vs. 1.8; 0.01; 0.63), CD8ab (0.48 vs. 0.53; 0.003; 0.72), CD4+/CD45RO+ (26.5 vs. 21.7; 0,001; 0,84), CD38ab (0.9 vs. 1.2; 0.005; 0.68), CD69ab (0.06 vs. 0.07; 0.029; 0.42), CD4+/CD45RA+ (0.41 vs. 0.54; 0.044; 0.49) and CD8+/CD38+ab (0.16 vs. 0.23; 0.003; 0.63).

Out of the changes in the course of five years, the change of CD3 was statistically significant at the 5% level of significance (absolute and relative). For groups of CSPs (A), CD3 decreased over five years; on the contrary, in the non-CSP groups, CD3 increased. For the CSP group, the mean absolute decrease was three units, and in the non CSP group, there was an increase of 2.2 (*p* value = 0.004; effect size = 0.73). Another significant change was for CD8. In the CSP group, there was a decrease (4.3 units, (11.7%)), and in the group non-CSP, there was an increase (one unit (4.8%).) The effect size was the highest of all parameters (0.88 for absolute change and 0.79 for relative change.) Other statistically significant differences were found in the changes in the characteristics: CD8ab, CD19, absolute total leukocyte count, CD3+/CD69+ab, CD5, CD40, CD69, CD69ab and CD8+/CD38+ab.

Changes and values obtained in the fifth year can be interpreted as significantly correlated with the CSP.

According to the effect size, the most significant correlation of the EDSS progression appears to be the change in CD8 (0.88), change in CD3 (0.73) and baseline value relative lymphocyte count (0.69), which can be thought of as the CSP predictor. In addition, the CD4+/CD45RO+ and CD3+/CD69+ab parameters are interesting: the groups differ both in the measured values and in their changes (Table 1). The descriptive statistics for all 37 parameters are summarized in Appendix A.

### 3.2. Validation of CSP Dependence on Significant Parameters after Correction of Influential Factors

The values of the lymphocyte characteristics at the baseline can be considered as potential biomarkers for MS if they can predict CSP with high reliability (accuracy). Using the logistic regression, we have estimated the predictive potential for each parameter. EDSS, age at baseline, and gender, based on our patient population, do not appear to be appropriate predictors (do not predict whether patient belongs has CSP; CSP = yes/no, pseudo R2 = 0; 0.02; 0).

The statistically significant parameters (baseline values of lymphocyte characteristics) and, therefore, potential biomarkers according to the pseudo R2 are: CD4+/CD45RO+ (0.31), CD3+/CD69+ab (0.27), CD69 (0.23), CD3+/CD69+ (0.19), CD19 (0.19), CD4+/CD45RA+ (0.18), CD4+/CD45RO+ab (0.18), relative lymphocyte count (0.15), and CD8 (0.07).

Except for CD19, the presented values of the lymphocyte characteristics relative to the baseline are statistically significant, even after adjusting for the influential factors EDSS, age, and sex.

Table 2 and Appendix A provide estimates of the odds ratio (OR) and pseudo R2 for all described parameters above (not only baseline) with ORs that are statistically significant without and/or with adjustment for influential factors.

### 3.3. Identification of Combinations of Explanatory Variables

#### 3.3.1. Penalized Logistic Regression (Identification of All 37 Explanatory Variables)

The lasso logistic regression was used to identify a significant combination of explanatory variables from all 37 explanatory variables. A combination of four parameters was selected at the baseline: relative lymphocyte counts, CD3+/CD69+, CD4+/CD45RO+, and CD4+/CD45RA+ab.

The selected four parameters were entered as explanatory variables into a multiple model. The value of R2 is 0.38. This model, which contains four lymphocyte parameters to baseline, fits the data by 23% (0.38/0.31) better than the best univariate parameter, CD4+/CD45RO+, at baseline. The statistically significant difference at the 5% level of significance explaining the parameter is CD4+/CD45RO+ (even after the influential factor adjustment). This finding confirms the results of the univariate analysis.

The estimated ORs and their statistical significance are provided in Table 3.

#### 3.3.2. Combination of Statistically Significant Variables from One-dimensional Models

The statistically significant baseline parameters in the one-dimensional regression models are: CD4+/CD45RO+, CD3+/CD69+ab, CD69, CD3+/CD69, CD19, CD4+/CD45RA+, CD4+/CD45RO+ab, relative lymphocyte count, and CD8.

Two parameters are strongly correlated: CD3+/CD69+ and CD3+/CD69+ab (Spearman’s correlation coefficient 0.92). In the multiple regression model, the CD3+/CD69+ab parameter was selected from this pair based on the higher R2 value of the one-dimensional models.

As an explanatory variable for the multiple regression model, a combination of the following eight parameters was selected: CD4+/CD45RO+, CD3+/CD69+ab, CD69, CD19, CD4+/CD45RA+, CD4+/CD45RO+ab, relative lymphocyte count, and CD8.

The R2 value of the multiple model R2 is 0.52. This model, which contains eight lymphocyte parameters to the baseline, fits the data 68% (0.52/0.31) better than the best univariate parameter at baseline CD4+/CD45RO+. The results of this model (regression parameter estimates) are not statistically significant due to the small number of patients in the groups and the complexity of the model, in which nine model parameters are estimated (intercept + eight for each lymphocyte parameter).

This combination of lymphocytic characteristics also fits the data better (R2 = 0.46 versus 0.38) than the combination of four variables described in the previous section.

The estimated ORs and their statistical significance are provided in Table 4.

The predicted probability of the CSP for the values of significant parameters, i.e., CD3+/CD69+ (at baseline), CD3+/CD69+ab (at baseline), CD4+/CD45RA+ (at baseline), CD4+/CD45R0+ (at baseline), CD4+/CD45RO+ab (at baseline), CD8 (at baseline), CD69 (at baseline), ad relative lymphocyte count (at baseline) is presented in Appendix A.

#### 3.3.3. ROC Analysis

Table 5 below presents the results of the ROC analysis for the four most appropriate parameters (lymphocyte characteristics).

The best discriminatory ability to distinguish the CSP patients from the non-CSP patients in our groups (sample) is illustrated by CD4+/CD45RO+ (AUC 76%). The higher the value of this parameter is, the more likely it is that the patient reaches CSP within five years. The cut-off for this parameter is 22.55 (accuracy 72.1%, sensitivity 70%, specificity 74%). The remaining parameters have a lower prediction potential (AUC 62–65%). Their cut-offs are provided in Table 5.

The results (the CSP prediction ability of identified cut-offs) were further validated in an independent cohort of patients treated with interferon β-1b (IFN). In Table 6, the original cut-offs (identified in the new, current data, i.e., patients treated with IFN) were verified. Table 7 summarizes the search for cut-off values for eight parameters in the independent set of patients treated with IFN.

## 4. Discussion

The objective of this study was to find a parameter or a combination of parameters that can predict the progression of disability in MS patients treated with GA (evaluated over a five-year period). In our study, using the ROC analysis, we determined cut-off values to the baseline that can determine the CSP and the non-CSP patients. The parameter, which helps best to identify the patients with an increased risk of disease progression within five years, is CD4+/CD45RO+.

For the CD4+/CD45RO+ characteristics, the cut-off value is set to baseline 22.55. If the patient has a value greater than 22.55, 70% of the patients can reliably be identified in the CSP group (sensitivity); in other words, 70% of the patients with a value >22.55 actually have CSP, and with a 74% probability, this cut-off can correctly identify the non-CSP patients (specificity).

We validated the ability of identified potential biomarkers and their cut-offs to predict CSP on an independent group of patients treated with IFN, i.e., with a drug with a different mechanism of action against GA, in order to assess the robustness of these cut-offs (general utilization for patients with MS independent of their treatment). When comparing the results, the CD4+/CD45RO+ > 22.55 cut-off retained a high sensitivity (72%) on the test file, i.e., it was able to identify correctly with this CSP validator group but failed to correctly identify the non-CSP patients (low specificity) (Figure 1). Other parameters did not reach satisfactory results.

The leucocyte molecule CD45 deserves special attention in the pathophysiology of MS. CD45 is a transmembrane molecule with tyrosine phosphatase activity, which is expressed at different density on all cells of haematopoietic origin, and exist in many different isoforms. Most naïve human T cells express a form of CD45R that is called CD45RA, and memory T cells express a different isoform called CD45RO. Several subtypes of memory T cells could be identified by mapping the expression of selected membrane molecules. Central memory T cells (TCM) express CD45RO, C-C chemokine receptor type 7 (CCR7), and L-selectin (CD62L) and are predominantly localized in lymphoid tissue. Effector memory T cells (TEM) that express CD45RO but lack expression of CCR7 and L-selectin are found both in the blood and tissues. Tissue resident memory T cells (TRM) occupy tissues without circulating in the blood.

The role of CD4+/CD45RO+ cells in the pathogenesis of MS is not yet fully understood. CD4+/CD45RO+ are memory helper T-lymphocytes, i.e., lymphocytes that have already been activated by antigens. This population changes with age, reaching a peak later in adulthood. In the experimental autoimmune encephalomyelitis, encephalitogenic T cells differ from the non-encephalitogenic ones by their expression of CD49d. The CD49d molecule is a β chain of the VLA4 integrin heterodimer adhesion molecule. The ligand for VLA4 adhesion molecule is VCAM1 (CD106) molecule expressed on endothelial cells of the brain-blood barrier. The interaction between VLA4 and VCAM1 molecules is a prerequisite for T cell entry from blood into the brain. The disease-inducing CD49d+(high) cells but not the CD49d+(low) cells enter the brain parenchyma. In this context, Barrau et al. characterized CD4+/CD45RO+/CD49d+(high) cells in RR MS patients. These cells, showing characteristics of activated cells able to produce proinflammatory cytokines, were found to be increased in the peripheral blood during relapses and present in high numbers in the cerebrospinal fluid. These results suggested that the CD4+/CD45RO+/CD49d+(high) subpopulation in RR MS patients includes autoreactive cells. CD4+/CD45RO+/CD49d+(high) cells show characteristics of activated T cells and are able to produce major TH1 cytokines such as IFN-γ [14].

CD4+/CD45RO+ memory T-cells from MS patients also showed a reduced ability to suppress NLRP3 inflammasome activation. NLRP3 inflammasome, which is assembled via both damage-associated molecular patterns (DAMPs) and pathogen-associated molecular patterns (PAMPs) is activating caspase 1. Caspase 1 in cooperation with several other molecules is able to cleave latent forms of proIL-1β and IL-18 cytokines. Fully active pluripotent proinflammatory cytokines IL-1β and IL-18 are produced in this way. NLRP3 inflammasomes exert key roles in the initiation and propagation of the inflammation [15].

As other potentially suitable biomarkers, we separately identified the relative lymphocyte count, CD3+/CD69+ and CD4+/CD45RA+ab or combined with CD4+/CD45RO+.

Blanco et al. and Pavelek et al. observed an increase in the CD4+/CD45RA+ count in GA-treatment responders. This observation is probably the result of the switch from CD4+/CD45RO+ memory T cells to naïve CD4+/CD45RA+ T cells, which is likely the result of GA action. The upregulation of CD4+/CD45RA+ appears to be one of the mechanisms by which GA inhibits MS activity [16,17]. CD3+/CD69+ cells represent early-activated T lymphocytes. A decrease in the CD3+/CD69+ count was also seen during the GA treatment [17].

According to the ROC analysis, the cut-off values for relative lymphocyte count are 26.2, for CD3+/CD69+ are 2.25 and for CD4+/CD45RA+ab are 0.435. The cut-off value of 26.2 relative lymphocyte count has a specificity of 71%. The sensitivity of this cut-off is low at 50%. The specificity of cut-off 2.25 for CD3+/CD69 + again has a low sensitivity of 53% and a high specificity of 76%. Conversely, the cut-off of 0.435 for CD4+/CD45RA+ab has a high sensitivity of 70% and a low specificity of 51%.

Although much is left to be clarified about pathogenic mechanisms of MS, understanding the mechanisms of the immunity-mediated damage to CNS components with MS enables the introduction of new medicines that positively modulate the damaging inflammation. With the development of the MS treatment, the early escalation of treatment has gained ground, which has given hope to patients in terms of curbing irreversible disease progression and maintaining a satisfactory quality of life. To achieve this goal, the monitoring of the clinical and subclinical disease activity is crucial.

Currently, there are several potential biomarkers, which are known as responders to the GA therapy. Despite this, none of these biomarkers have been introduced in common clinical practice. A high IL-18 level at baseline and a reduction in TNF-alpha over time are associated with the response to GA [18]. Regarding the association with clinical responders to the GA treatment, Mindur et al. [19] reported an increase in serum IL-27 production. Another study found increased expression of the Response Gene to Complement 32 (RGC-32) (*p* < 0.0001) and FasL (Fas ligand, CD178) (*p* < 0.0001) and decreased expression of IL-21 (*p* = 0.02) [20] as potential biomarkers. RGC-32 was also detected by Tatomir et al. as a potential biomarker of relapse and response to the GA therapy, as the RGC-32 mRNA expression is significantly decreased during relapse and increased in responders to the GA treatment. The predictive accuracy of RGC-32 as a potential biomarker (by ROC analysis) is 90% for detecting relapses and 85% for detecting a response to the GA treatment [21].

Other biomarkers for responders to the GA treatment may be SIRT1 mRNA (a NAD-dependent histone and protein deacetylase) and H3K9me2 (H3K9 dimethylation). GA responders had significantly higher SIRT1 mRNA (*p* = 0.01) and H3K9me2 levels than non-responders (*p* = 0.018) [22]. The results were further evaluated by Cieriello et al. Statistically significant lower levels of p-SIRT1 protein (*p* = 0.02) and H3K9me3 (*p* = 0.004) were found in GA non-responders compared to the responders. Non-responders to the GA treatment were defined as patients who exhibited at least two relapses following initiation of the GA treatment. Using the receiver operating characteristic analysis, the area under the curve (AUC) for the prediction of relapse was 77% (*p* = 0.007) for p-SIRT1 and was 81% (*p* = 0.002) for H3K9me3. For predicting responsiveness to the GA treatment, the AUC was 75% (*p* = 0.01) for H3K9me3. H3K9me3 could serve as potential biomarker to predict response to the GA treatment [23].

Furthermore, patterns of TH1/TH2 cytokines can predict clinical response in MS patients treated with GA. The quotient (IL-2 + IFN-γ)/(IL-4 + IL-10) was elevated in patients with relapses compared to relapse-free patients after 12 (*p* = 0.04), 24 (*p* = 0.02) and 36 months (*p* = 0.04) [24].

This present study has some limitations. Pregnancy and compliance rate were not observed, and the results can be influenced by these factors. Although a worsening of the neurological findings measured by the EDSS scale was found in a number of patients over the five-year period, these patients did not meet the criteria for escalation of treatment or SPMS [25]. As they were patients in routine clinical practice, escalation was tied to valid local Czech insurance reimbursement criteria for treatment.

The disadvantage of peripheral blood biomarkers is that they are released more extracerebrally than directly from the brain [26]. However, the collection of cerebrospinal fluid to evaluate biomarkers is a relatively invasive method with more frequent adverse effects. In contrast, peripheral blood collection and subsequent analysis is a quick and simple method. We are aware that absolute lymphocyte counts are highly variable and may even change considerably during a day. Therefore, peripheral blood collection was performed in the morning at approximately the same time. We believe that analysing the CD4+/CD45RO+ parameter and finding baseline values before starting MS treatment can help doctors consider the use of GA in common clinical practice.

## 5. Conclusions

From the univariate and multivariate regression analyses, CD4+/CD45RO+ lymphocytes best describe the CSP dependence on lymphocyte parameters in our cohort. In addition, the following promising biomarkers seem worth examining: relative lymphocyte count, CD3+/CD69+ for early activated T cells and CD4+/CD45RA+ for naïve T cells, which can be considered either alone or in combination with CD4+/CD45RO+ memory T cells. The analysis of the difference between groups confirms that strong differences exist in these parameters. Furthermore, the groups differ strongly in the change of CD8 and CD3 T cell populations. The correlation analysis confirms our findings from regression modelling. The CD4+/CD45RO+ parameter was also identified by the ROC analysis as a potential biomarker, and a measured value above 23 means an increased risk of disease progression within five years.

## Figures and Tables

**Figure 1 cells-08-00456-f001:**
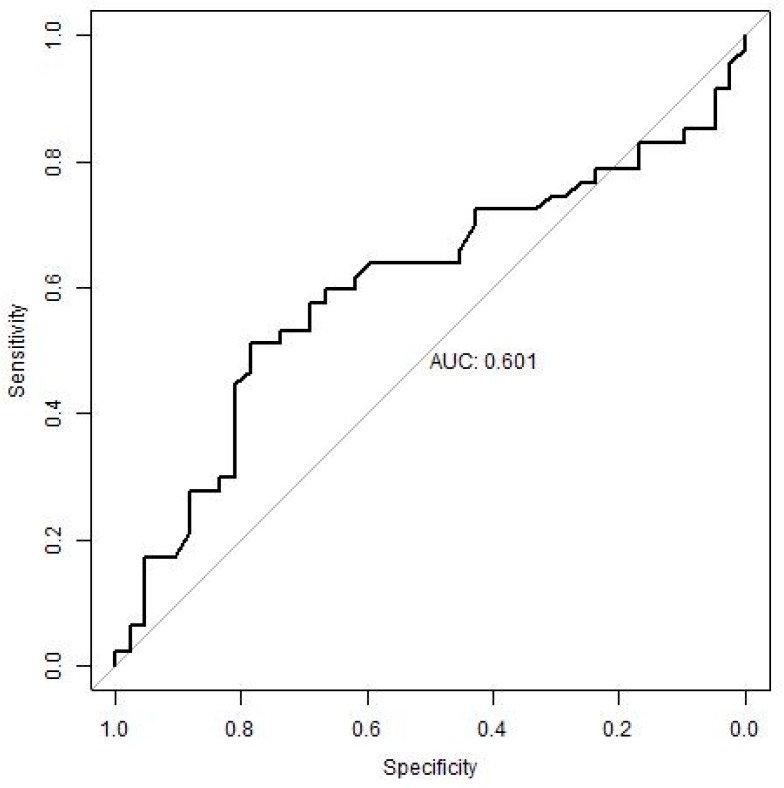
ROC curve for CD4+/CD45RO+ at baseline in the validation group (IFN). Legend: The receiver-operating characteristic (ROC) curve (in bold) for the CD4+/CD45RO+ lymphocytes is shown. This lymphocyte parameter performed best to distinguish between CSP and non-CSP patients in this study. The presented curve was estimated on independent (test) data.

**Table 1 cells-08-00456-t001:** Descriptive characteristics and EDSS progression groups.

Characteristic		All (72)	CSP (32)	Non-CSP (40)	Groups’ Comparison	
					*p* Value	Effect Size
EDSS	Baseline	2.40 ± 1.35 (2.00)	2.42 ± 1.13 (2.00)	2.38 ± 1.51 (1.5)	0.438	0.03
	5 Years	3.31 ± 1.7 (3.00)	4.22 ± 1.37 (4.00)	2.58 ± 1.59 (2.00)	<0.001	1.1
	abs.	0.91 ± 1.01 (0.50)	1.80 ± 0.90 (1.50)	0.20 ± 0.25 (0.00)	<0.001	2.56
	rel. (%)	47.43 ± 59.16 (28.57)	92.35 ± 62.01 (87.50)	11.50 ± 18.30 (0.00)	<0.001	1.86
CD3	Baseline	76.27 ± 7.19	77.92 ± 7.09	74.96 ± 7.08	0.073	0.42
	5 Years	76.15 ± 7.02	74.94 ± 6.88	77.13 ± 7.06	0.19	0.31
	abs.	−0.12 ± 7.45	−2.98 ± 7.97	2.17 ± 6.20	0.004	0.73
	rel. (%)	0.35 ± 9.85	−3.29 ± 10.15	3.27 ± 8.67	0.004	0.7
CD8	Baseline	26.9 ± 6.52	28.54 ± 7.50	25.59 ± 5.36	0.055	0.46
	5 Years	25.51 ± 5.93	24.21 ± 4.96	26.55 ± 6.48	0.096	0.4
	abs.	−1.39 ± 6.52	−4.33 ± 6.93	0.97 ± 5.14	<0.001	0.88
	rel. (%)	−2.53 ± 22.5	−11.74 ± 20.64	4.84 ± 21.38	0.001	0.79
Relative lymphocyte count	Baseline	29.45 ± 8.68	26.28 ± 7.91	31.98 ± 8.52	0.005	0.69
	5 Years	30.94 ± 7.78	29.16 ± 8.52	32.37 ± 6.92	0.082	0.42
	abs.	1.49 ± 9.9	2.88 ± 11.31	0.39 ± 8.60	0.292	0.25
	rel. (%)	13.27 ± 44.27	22.65 ± 56.16	5.77 ± 30.52	0.421	0.39
CD4+/CD45RO+	Baseline	22.2 ± 6.37	25.13 ± 5.88	19.88 ± 5.83	<0.001	0.9
	5 Years	23.79 ± 6.11	26.45 ± 5.39	21.66 ± 5.87	0.001	0.84
	abs.	1.46 ± 5.21	1.33 ± 6.15	1.56 ± 4.42	0.86	0.04
	rel. (%)	10.05 ± 25.53	8.74 ± 25.75	11.09 ± 25.65	0.709	0.09
CD3+/CD69+ab	Baseline	0.06 ± 0.06	0.07 ± 0.05	0.05 ± 0.06	0.043	0.32
	5 Years	0.03 ± 0.02	0.03 ± 0.01	0.03 ± 0.02	0.061	0.41
	abs.	−0.03 ± 0.06	−0.05 ± 0.05	−0.02 ± 0.06	0.004	0.43
	rel. (%)	−11.63 ± 78.81	−39.76 ± 59.21	10.42 ± 85.74	0.003	0.67

Legend: The reported statistics are mean ± standard deviation, except for gender and EDSS (expanded disability status scale). Gender is summarized according to the frequency and proportion of females. The median is added (in brackets) to the mean and standard deviation of EDSS due to its positively skewed distribution. The effect size is assessed by Cohen’s D. All *p* values are reported after the Benjamini-Hochberg correction.

**Table 2 cells-08-00456-t002:** Univariate logistic regression models for baseline values of lymphocyte parameters and multivariate models with adjustment by the covariates EDSS, age and gender.

Variable	Univariate Logistic Regression					Multivariate Logistic Regression with Covariates				
	OR	95% LCI OR	95% UCI OR	*p*	R2	OR	95% LCI OR	95% UCI OR	*p*	R2
EDSS	1.03	0.72	1.46	0.8826	<0.001	0.92	0.61	1.38	0.7008	0.02
age	1.03	0.98	1.09	0.306	0.02					
sex	1.31	0.37	4.64	0.672	<0.001					
CD8	1.08	1	1.17	0.0607	0.07	1.08	1	1.18	0.0477	0.1
CD19	0.87	0.76	0.98	0.0367	0.19	0.88	0.76	1	0.0601	0.21
Relative lymphocyte count	0.92	0.85	0.97	0.008	0.15	0.9	0.83	0.97	0.0077	0.17
CD3+/CD69+	1.3	1.04	1.69	0.0297	0.19	1.31	1.04	1.71	0.0296	0.23
CD69	1.27	1.05	1.57	0.0178	0.23	1.26	1.04	1.56	0.0265	0.25
CD4+/CD45RO+	1.17	1.07	1.31	0.002	0.31	1.17	1.07	1.32	0.0027	0.32
CD4+/CD45RA+	0.93	0.87	0.99	0.0359	0.18	0.93	0.87	1	0.0481	0.21
CD3+/CD69+ab	1341165	9.63	1.19E+12	0.0276	0.27	1454961	8.83	1.36E+12	0.0277	0.29
CD4+/CD45RO+ab	37.95	1.42	1536.95	0.0386	0.18	41.79	1.2	2304.94	0.0494	0.21

Legend: OR, odds ratio; 95% LCI OR, lower 95% confidence limit of OR; 95% UCI, upper 95% confidence limit of OR; R2, Nagelkerke’s pseudo R squared to assess the quality of fit (the higher, the better).

**Table 3 cells-08-00456-t003:** Multivariate logistic regression model and its adjustment by the covariates EDSS, age and gender.

Variable	Univariate Logistic Regression					Multivariate Logistic Regression with Covariates				
	OR	95% LCI OR	95% UCI OR	*p*	R2	OR	95% LCI OR	95% UCI OR	*p*	R2
(Intercept)	0.10	0.00	4.53	0.24	0.38	0.07	0.00	10.68	0.32	0.39
Relative lymphocyte count	0.96	0.88	1.04	0.33		0.95	0.86	1.04	0.31	
CD3+/CD69+	1.28	0.99	1.72	0.07		1.28	0.98	1.72	0.08	
CD4+/CD45RO+	1.12	1.01	1.27	0.04		1.13	1.02	1.29	0.03	
CD4+/CD45RA+ab	0.95	0.04	26.11	0.98		1.10	0.04	34.99	0.95	

Legend: OR, odds ratio; 95% LCI OR, lower 95% confidence limit of OR; 95% UCI, upper 95% confidence limit of OR; R2, Nagelkerke’s pseudo R squared to assess the quality of fit (the higher, the better).

**Table 4 cells-08-00456-t004:** The second multivariate logistic regression model and its adjustment by covariates: EDSS, age and gender.

Variable	Multivariate Logistic Regression					Multivariate Logistic Regression after Adjustment				
	OR	95% LCI OR	95% UCI OR	*p*	R2	OR	95% LCI OR	95% UCI OR	*p*	R2
(Intercept)	0.58	0	614.57	0.877	0.46	0.69	0	3074.75	0.929	0.54
CD8	1.03	0.92	1.16	0.6		1.03	0.92	1.16	0.6	
CD19	0.92	0.78	1.06	0.254		0.92	0.77	1.06	0.269	
Relative lymphocyte count	0.95	0.86	1.04	0.256		0.94	0.83	1.04	0.219	
CD3+/CD69+	1.9	0.56	7.44	0.321		1.97	0.58	7.69	0.295	
CD69	1.03	0.68	1.55	0.875		0.98	0.63	1.49	0.92	
CD4+/CD45RO+	1.01	0.82	1.26	0.907		1.02	0.83	1.26	0.869	
CD4+/CD45RA+	0.95	0.86	1.04	0.262		0.94	0.85	1.04	0.238	
CD3+/CD69+ab	0	0	1.67E+16	0.534		0	0	2.29E+16	0.542	
CD4+/CD45RO+ab	74.45	0.02	1474731	0.341		125.86	0.03	1630320	0.272	

Legend: OR, odds ratio; 95% LCI OR, lower 95% confidence limit of OR; 95% UCI, upper 95% confidence limit of OR; R2, Nagelkerke’s pseudo R squared to assess the quality of fit (the higher, the better).

**Table 5 cells-08-00456-t005:** Discriminative ability of cut-offs of parameters to correctly distinguish between CSP and non-CSP patients.

Parameter	AUC	Cut-off	Sensitivity	Specificity	Accuracy
Relative lymphocyte count	64%	<26.2	50%	71%	62%
CD3+/CD69+	65%	>2.25	53%	76%	66%
CD4+/CD45RO+	76%	>22.55	70%	74%	72.1%
CD4+/CD45RA+ab	62%	<0.435	70%	51%	59.7%

**Table 6 cells-08-00456-t006:** Validation of the predictive ability of the identified cut-offs on an independent data set (IFN).

Parameter	AUC	Cut-off	Old Cut-offs Verified on a New Independent Data Set (Testing Data Set—IFN Data)		
			Sensitivity	Specificity	Accuracy
Relative lymphocyte count	58%	<26.20	49%	77%	62%
CD3+/CD69+	53%	>2.25	72%	39%	57%
CD4+/CD45RO+	60%	>22.55	72%	36%	56%
CD4+/CD45RA+ab	56%	<0.44	72%	43%	59%

**Table 7 cells-08-00456-t007:** Identification of new cut-offs in the interferon data (IFN) data.

Parameter	AUC	Cut-off	Sensitivity	Specificity	Accuracy
Relative lymphocyte count	58%	<23.50	38%	86%	61%
CD3+/CD69+	53%	>2.55	70%	43%	57%
CD4+/CD45RO+	60%	>26.00	60%	67%	62.9%
CD4+/CD45RA+ab	56%	<0.45	74%	43%	59.6%
CD8	56%	>25.30	60%	55%	57%
CD69	52%	>3.15	83%	34%	60%
CD38	48%	<54.00	51%	57%	54%
CD3+/CD25+	48%	<4.95	49%	64%	56%

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
