# Peer review of "CD4+/CD45RO+: A Potential Biomarker of the Clinical Response to Glatiramer Acetate"

_cells, 2019, doi:10.3390/cells8050456_

Reviewer 1 Report

Work in this interesting manuscript examined individual populations of lymphocytes in two groups of multiple sclerosis (MS) patients - clinically isolated syndrome (CIS) and relapsing-remitting MS (RR-MS)- after five years of treatment with glatiramer acetate. Their findings revealed that CD4+/CD45RO+ could be used as a biomarker to detect patients with confirmed sustained progression of the disease. The work includes an extensive and thorough analysis of lymphocyte populations and careful statistical analysis of their association to the patient groups. The findings are highly important for the evaluation of disease progression in MS.

Minor points:

The definition for the abbreviation EDSS, expanded disability status scale, appears to be missing in the text. 

Also missing are the legends to the figures.  Perhaps Figures 1-8 could be combined in a single figure with a shared legend.

Author Response

Hradec Kralove,

Czech Republic

March 25,  2019

Dear Reviewer,

We are writing to you about our article called CD4+/CD45RO+: potential biomarker of clinical response to glatiramer acetate, which needed revisions in order to be considered for the publication in this journal.

We thank you for your constructive and justified comments. We tried hard to incorporate the comments (consult below) in a very thorough manner. Overall, thanks to the review, the article has been improved.

Yours sincerely,

                        ZbyÅ¡ek Pavelek

Department of Neurology

Comments and Suggestions for Authors

Work in this interesting manuscript examined individual populations of lymphocytes in two groups of multiple sclerosis (MS) patients - clinically isolated syndrome (CIS) and relapsing-remitting MS (RR-MS)- after five years of treatment with glatiramer acetate. Their findings revealed that CD4+/CD45RO+ could be used as a biomarker to detect patients with confirmed sustained progression of the disease. The work includes an extensive and thorough analysis of lymphocyte populations and careful statistical analysis of their association to the patient groups. The findings are highly important for the evaluation of disease progression in MS.

Minor points:

The definition for the abbreviation EDSS, expanded disability status scale, appears to be missing in the text. 

Definition of EDSS was added to the text.

Also missing are the legends to the figures.  Perhaps Figures 1-8 could be combined in a single figure with a shared legend.

Certainly, we agree, that a legend explaining figures need to be specified. We would like to add the common legend for Figures 1-8 as follows:

Legend: The figure shows the predicted probability of the CSP (on vertical axis) for the range of measured values (on the horizontal axis) of a lymphocyte parameter (labelled at the bottom of the figure). The blue curve depicts the point estimate of the probability and pink area represents its 95% confidence interval.

We prefer a single figures than a combined one due to the readability (clarity).

Please, add also legend for Figure 9 as follows:

Legend: The figure shows the ROC curve (in bold) for the CD4+/CD45RO+. This lymphocyte parameter performed the best to distinguish between CSP and non CSP patient in this study. The presented curve was estimated on independent (test) data

Reviewer 2 Report

The paper by Valis et al, describes potential biomarkers for the earliest stages of relapse remitting (RR) multiple sclerosis (MS). I found this article interesting to read however it also raised a few concerns.

MAJOR COMMENTS

Methods section

This article measured peripheral blood samples in 12 male and 60 female participants before and again 5 years after treatment of subcutaneous injection (20mg/day) with GA.  Was this daily treatment continuous? Did the authors obtain a compliance rate of the participants for this treatment over 5 years?  Could the results be different if doses were missed?  Since RR MS stays the same or improves during pregnancy, did any female participant stop taking the treatment because of pregnancy?  I feel the results of this study may be different if any of these occurred during the 5 years of the study. If this information was not obtained, then state it as a limitation to the study.

The validation group for the ROC analysis also measured peripheral blood samples from 25 male and 72 female participants before and again 5 years after treatment of interferon beta 1-b.  What is the dosing for this treatment? Was this treatment also a daily/weekly/monthly injection? If not daily, then the results may be confounding and should be stated as a limitation to the study. What was the compliance rate over the 5 years of the study given what was said in #1 above.

Line 116-118: More information is needed to describe what and how were the 37 parameters selected to compare the CSP to the non-CSP cohort. 

Results section

Line 168-169 says "In group A there are 26 out of 32 women......Group B has 34 women out of 40....."  Did these 72 women come from the ROC analysis who were taking interferon beta 1-b treatment?  If these 72 women are taking interferon beta 1-b, where is the breakdown of CSP and non-CSP women taking GA? More information is needed in the methods section on selecting the Group A (CSP) and Group B (non-CSP) cohort to address these concerns for this analysis. 

If the 72 women are only taking interferon beta 1-b and this study was identifying a potential biomarker of clinical response to GA, why start showing the results for the 72 women taking interferon beta 1-b treatment in Table 1? Perhaps reword the title for this paper or drop Tables 1-4 if just showing interferon beta 1-b results. Were the men ever included in this analysis?  If so, then consider redefining your cohorts of 72 participants as it's difficult to determine if these are GA or Interferon beta 1-b treated individuals.

Was the 60 females taking the GA treatment ever selected for analysis in this study?  Consider adding the breakdown of participants analyzed in Tables 2-5 to make it easier to understand the cohort selected for the analysis. Similarly, consider adding the participants analyzed in the Figures 1-9. 

MINOR COMMENTS

Line 189 says "In the fifth year, the groups differ statistically significantly in absolute lymphocyte counts...". Perhaps consider changing this to read " In the fifth year, the differences in the groups were statistically significant in absolute lymphocyte counts...."

What are Non SP (40) participants in the Table 1 heading?  Should this read Non CSP (40)?

 In the Results section, what is the significance of having some bolded variables in Table 2 and bolded baseline values in Table 3? Similarly, what is the significance of bolding (0.44) on Line 230?

Author Response

Hradec Kralove,

Czech Republic

March 25,  2019

Dear Reviewer,

We are writing to you about our article called CD4+/CD45RO+: potential biomarker of clinical response to glatiramer acetate, which needed revisions in order to be considered for the publication in this journal.

We thank you for your constructive and justified comments. We tried hard to incorporate the comments (consult below) in a very thorough manner. Overall, thanks to your review, the article has been improved.

Yours sincerely,

                        ZbyÅ¡ek Pavelek

Department of Neurology

Methods section

This article measured peripheral blood samples in 12 male and 60 female participants before and again 5 years after treatment of subcutaneous injection (20mg/day) with GA.  Was this daily treatment continuous? Did the authors obtain a compliance rate of the participants for this treatment over 5 years?  Could the results be different if doses were missed?  Since RR MS stays the same or improves during pregnancy, did any female participant stop taking the treatment because of pregnancy?  I feel the results of this study may be different if any of these occurred during the 5 years of the study. If this information was not obtained, then state it as a limitation to the study.

All patients were treated with GA (subcutaneous injection, 20 mg each day). Compliance rate and pregnancy were not monitored in patients. These facts were explained in the part on Discussion as limitations to this study.

The validation group for the ROC analysis also measured peripheral blood samples from 25 male and 72 female participants before and again 5 years after treatment of interferon beta 1-b.  What is the dosing for this treatment? Was this treatment also a daily/weekly/monthly injection? If not daily, then the results may be confounding and should be stated as a limitation to the study. What was the compliance rate over the 5 years of the study given what was said in #1 above.

All patients were treated with INFβ-1b. INFβ-1b (250 µg) was administered via subcutaneous injection every other day. Compliance was not monitored in patients. This limitation has been explained in the part on Discussion.

Line 116-118: More information is needed to describe what and how were the 37 parameters selected to compare the CSP to the non-CSP cohort. 

Multiple peripheral blood parameters were assessed in absolute and relative values. These parameters included lymphocytes, CD4+ T-lymphocytes, CD8+ T-lymphocytes, CD 19 (B-lymphocytes), natural killer cells (CD3-/CD16+56+), CD3+/CD69+ cells, CD5 cells, CD25 cells, CD3+/CD25+ cells, CD5+/CD19+ cells, CD4+/CD45RO+ cells, CD8+/CD38+ cells, CD4+/CD45RA+ cells, CD69 protein, CD40 protein, and CD40L protein. Additionally, the absolute leukocyte (white blood cell) count was determined. This is a total of 37 parameters and this fact has been incorporated into the manuscript.

Results section

Line 168-169 says "In group A there are 26 out of 32 women......Group B has 34 women out of 40....."  Did these 72 women come from the ROC analysis who were taking interferon beta 1-b treatment?  If these 72 women are taking interferon beta 1-b, where is the breakdown of CSP and non-CSP women taking GA? More information is needed in the methods section on selecting the Group A (CSP) and Group B (non-CSP) cohort to address these concerns for this analysis. 

The group of patients treated with GA consisted of 72 patients (12 male patients and 60 female patients). Group A (CSP) consisted of 32 patients (6 male patients and 26 female patients), group B consisted of 40 patients (6 male patients and 34 female patients). In total, it makes 72 patients described above, whose peripheral blood results were further statistically analyzed.  

The group of patients treated with INFβ- 1b consisted of 97 patients (25 male patients and 72 female patients) and represented a validation group for the ROC analysis.

If the 72 women are only taking interferon beta 1-b and this study was identifying a potential biomarker of clinical response to GA, why start showing the results for the 72 women taking interferon beta 1-b treatment in Table 1? Perhaps reword the title for this paper or drop Tables 1-4 if just showing interferon beta 1-b results. Were the men ever included in this analysis?  If so, then consider redefining your cohorts of 72 participants as it's difficult to determine if these are GA or Interferon beta 1-b treated individuals.

Table 1 provides the results for 72 patients treated with GA (12 male patients and 60 female patients)

Was the 60 females taking the GA treatment ever selected for analysis in this study?  Consider adding the breakdown of participants analyzed in Tables 2-5 to make it easier to understand the cohort selected for the analysis. Similarly, consider adding the participants analyzed in the Figures 1-9. 

Yes, they were. 60 females treated with GA were analyzed in this study. 72 females treated with INFβ-1b were analyzed within the ROC analysis as a validation group.

MINOR COMMENTS

Line 189 says "In the fifth year, the groups differ statistically significantly in absolute lymphocyte counts...". Perhaps consider changing this to read " In the fifth year, the differences in the groups were statistically significant in absolute lymphocyte counts...."

This sentence was modified accordingly. Please consult the manuscript.

What are Non SP (40) participants in the Table 1 heading?  Should this read Non CSP (40)?

This mistake was corrected; it should read as Non CSP. Please consult the manuscript.

 In the Results section, what is the significance of having some bolded variables in Table 2 and bolded baseline values in Table 3? Similarly, what is the significance of bolding (0.44) on Line 230?

Table 3 was deleted

Reviewer 3 Report

Critiques,

In the "Abstract" section the results are presented in one sentence. Results presentation should be extended. The Material and Methods section of the abstract should be reduced.

In the Material and methods sections replace "Data analysis" with "FACS analysis"

All the antibodies used, their provenience and dilution used should be introduced in a separate Table.

The legend of the Figures is very short. All the Legends need to be self explanatory.

Many if the Tables are very long and hard to read. They  need to be reduced and only results which are statistical significant.

Is unclear to me why the ROC data in Figure 9 are the only one which were chosen to be  presented. Other data with a better AUC should be used.

Author Response

Hradec Kralove,

Czech Republic

March 25,  2019

Dear Reviewer,

We are writing to you about our article called CD4+/CD45RO+: potential biomarker of clinical response to glatiramer acetate, which needed revisions in order to be considered for the publication in this journal.

We thank you for your constructive and justified comments. We tried hard to incorporate the comments (consult below) in a very thorough manner. Overall, thanks to your review, the article has been improved.

Yours sincerely,

                        ZbyÅ¡ek Pavelek

Department of Neurology

Comments and Suggestions for Authors

Critiques,

In the "Abstract" section the results are presented in one sentence. Results presentation should be extended. The Material and Methods section of the abstract should be reduced.

Thank you for this suggestion. We reduced details about the statistical analysis, which is explained in detail in the main text (Section Data Analysis).

In the Material and methods sections replace "Data analysis" with "FACS analysis"

This was modified accordingly. Please consult the manuscript.

All the antibodies used, their provenience and dilution used should be introduced in a separate Table.

Supplementary Table 1 was made and all the antibodies used, their provenience and dilution are provided.

The legend of the Figures is very short. All the Legends need to be self explanatory.

Certainly, we agree that legends explaining the figures need to be specified. We would like to add a common legend for Figures 1-8 as follows:

Legend: The figure shows the predicted probability of the CSP (on vertical axis) for the range of measured values (on the horizontal axis) of a lymphocyte parameter (labelled at the bottom of the figure). The blue curve depicts the point estimate of the probability and the pink area represents its 95% confidence interval.

We prefer a single figure than a combined one due to the overall clarity of each figure.

Please, add also legend for Figure 9 as follows:

Legend: The figure shows the ROC curve (in bold) for the CD4+/CD45RO+. This lymphocyte parameter performed the best to distinguish between CSP and non CSP patient in this study. The presented curve was estimated on independent (test) data.

Many if the Tables are very long and hard to read. They need to be reduced and only results which are statistical significant.

To make the manuscript more transparent, Table 2 was divided into two parts: Table 2 (for baseline values, which were further analyzed as potential biomarkers) and the Supplementary table for the remaining values (in the 5th year, absolute and relative changes in 5 years) to preserve the transparency and repeatability of this study. At the same time, the section concerning the Mutual relationships among the parameters (by explanatory variables) and EDSS (Correlation Analysis), including Table 3, was deleted.

Is unclear to me why the ROC data in Figure 9 are the only one which were chosen to be  presented. Other data with a better AUC should be used.

We presented the ROC curve for the CD4+/CD45RO+ because it classified best in both the original data (results in Table 6) and validation data (Table 7).

Round  2

Reviewer 2 Report

I thank the authors for their comments and improvements to their article however some clarification is still needed.

1. The validation group for the ROC analysis also measured peripheral blood samples from 25 male and 72 female participants before and again 5 years after treatment of interferon beta 1-b.  What is the dosing for this treatment? Was this treatment also a daily/weekly/monthly injection? If not daily, then the results may be confounding and should be stated as a limitation to the study. What was the compliance rate over the 5 years of the study given what was said in #1 above.

All patients were treated with INFβ-1b. INFβ-1b (250 µg) was administered via subcutaneous injection every other day. Compliance was not monitored in patients. This limitation has been explained in the part on Discussion.

Additional Comment: Please add the treatment rate of interferon beta 1-b being every other day to the manuscript. Also comment on how this cohort is useful for a validation group for the ROC study when investigating a daily injection treatment of GA. 

2. Line 168-169 says "In group A there are 26 out of 32 women......Group B has 34 women out of 40....."  Did these 72 women come from the ROC analysis who were taking interferon beta 1-b treatment?  If these 72 women are taking interferon beta 1-b, where is the breakdown of CSP and non-CSP women taking GA? More information is needed in the methods section on selecting the Group A (CSP) and Group B (non-CSP) cohort to address these concerns for this analysis. 

The group of patients treated with GA consisted of 72 patients (12 male patients and 60 female patients). Group A (CSP) consisted of 32 patients (6 male patients and 26 female patients), group B consisted of 40 patients (6 male patients and 34 female patients). In total, it makes 72 patients described above, whose peripheral blood results were further statistically analyzed.  

The group of patients treated with INFβ- 1b consisted of 97 patients (25 male patients and 72 female patients) and represented a validation group for the ROC analysis.

Additional Comment: Please consider adding a sentence to describe the type of patients selected to be divided into two groups (Group A (CSP) and Group B (CSP) since this sentence comes directly after talking about the ROC study in the results section.  It would be easier to the reader in the Statistical Processing section and in the Results section to know what sample cohort was selected when discussing the two groups. 

3.What are Non SP (40) participants in the Table 1 heading?  Should this read Non CSP (40)?

This mistake was corrected; it should read as Non CSP. Please consult the manuscript.

Additional Comment: I consulted the manuscript and the mistake still remains in the column title for Table 1.  Please consider correcting this mistake.

Author Response

Hradec Kralove,

Czech Republic

April 11, 2019

Dear Reviewer,

We are writing to you about our article called CD4+/CD45RO+: potential biomarker of clinical response to glatiramer acetate, which needed revisions in order to be considered for the publication in this journal.

We thank you again for your constructive and justified comments. We incorporated the comments (consult below) in a very thorough manner. New changes are highlighted in the manuscript in red. Overall, thanks to your review, the article has been improved.

Yours sincerely,

                        ZbyÅ¡ek Pavelek

Department of Neurology

Additional Comment: Please add the treatment rate of interferon beta 1-b being every other day to the manuscript. Also comment on how this cohort is useful for a validation group for the ROC study when investigating a daily injection treatment of GA.

Treatment rate was included. Please consult the manuscript.

We validated the ability of identified cut-offs to predict CSP on an independent group of patients treated differently (IFN) in order to assess the robustness of these cut-offs (general utilization for patients with MS independent to their treatment). This text has been included into the part on Discussion.

Please consider adding a sentence to describe the type of patients selected to be divided into two groups (Group A (CSP) and Group B (CSP) since this sentence comes directly after talking about the ROC study in the results section.  It would be easier to the reader in the Statistical Processing section and in the Results section to know what sample cohort was selected when discussing the two groups.

In the manuscript, this section has been modified for the number of patients and their division into groups.

What are Non SP (40) participants in the Table 1 heading?  Should this read Non CSP (40)?

This mistake was corrected; it should read as Non CSP. Please consult the manuscript.

Reviewer 3 Report

Acceptable for publication

Author Response

Hradec Kralove,

Czech Republic

April 11, 2019

Dear Reviewer,

We are writing to you about our article called CD4+/CD45RO+: potential biomarker of clinical response to glatiramer acetate, which needed revisions in order to be considered for the publication in this journal.

We thank you again for your constructive and justified comments. Overall, thanks to your review, the article has been improved.

Yours sincerely,

                        ZbyÅ¡ek Pavelek

Department of Neurology